# Same Day Discharge versus Inpatient Surgery for Robot-Assisted Radical Prostatectomy: A Comparative Study

**DOI:** 10.3390/jcm10040661

**Published:** 2021-02-09

**Authors:** Razvan George Rahota, Ambroise Salin, Jean Romain Gautier, Christophe Almeras, Guillaume Loison, Christophe Tollon, Jean Baptiste Beauval, Guillaume Ploussard

**Affiliations:** Department of Urology, La Croix du Sud Hospital, 31130 Quint Fonsegrives, France; ambroise.salin@gmail.com (A.S.); gautierjr@hotmail.fr (J.R.G.); c.almeras@gmail.com (C.A.); guillaumeloison@gmail.com (G.L.); tol@club-internet.fr (C.T.); jbbeauval@gmail.com (J.B.B.); g.ploussard@gmail.com (G.P.)

**Keywords:** prostate cancer, radical prostatectomy, enhanced recovery after surgery, outpatient, robot

## Abstract

(1) Background: no study has compared outcomes of same day discharge (SDD) versus inpatient robot-assisted radical prostatectomy (RARP) in homogenous cohorts. Our aim was to compare perioperative outcomes and urinary continence recovery between SDD and inpatient RARP in contemporary, comparable patients. (2) Methods: we included consecutive patients undergoing RARP between 2018 and 2020 (*n =* 376). Only patients eligible for SDD (no oral anticoagulant, distance home-hospital <150 km) and having >6-month follow-up were included (*n* = 180). All patients underwent RARP with or without lymph node dissection. Comparisons were performed between SDD (*n =* 42) and inpatient RARP (*n =* 138). Primary outcomes were 90-day complication and readmission rates and continence rates at 1 and 6 months. (3) Results: median patient age was 66.7 years. Median duration of surgery and blood loss was 134 min and 200 mL, respectively. Lymph node dissection and nerve-sparing procedures were performed in 76.7% and 82.2% of cases, respectively. Median follow-up was 19.5 months. No difference was seen regarding patient features, peri-operative outcomes, and pathology parameters between both groups. The proportion of SDD RARP was stable over time (23.5%). The 90-day unplanned visits, readmission and complication rates were 9.5%, 7.1%, and 19.0% in SDD patients versus 14.5% (*p* = 0.407), 10.1% (*p* = 0.560), 28.3% (*p* = 0.234) for inpatient RARP, respectively. Trends favoring SDD were not statistically significant. Continence rates at 1-(*p* = 0.589) and 6-months (*p* = 0.674) were comparable between SDD and inpatient RARP. The main limitation was the lack of randomization. (4) Conclusions: this multi-surgeon comparative study confirms the safety of routine SDD RARP in terms of perioperative and functional outcomes. Trends favoring SDD in terms of complications, emergency visits and readmission have to be confirmed.

## 1. Introduction

Robot-assisted radical prostatectomy is currently the preferred surgical approach for prostate cancer surgery and a reference treatment option for localized intermediate and high-risk prostate cancer [1,2]. The advent of robotic assistance has confirmed the benefits of minimally invasive surgery in terms of blood loss, pain, length of stay, and global post-operative recovery [3]. In addition, enhanced recovery after surgery (ERAS) regimens have led to continuous improvements regarding peri-operative outcomes in onco-urology surgery, even after robot-assisted radical prostatectomy (RARP) [4,5,6,7,8]. Recently, prehabilitation pathway along with ERAS and robotic surgery has also been suggested to synergistically participate in improving patient perception and post-operative outcomes after RARP [9]. All of these improvements may generate a reduction of length of stay and a wider acceptance of same day surgery (SDD).

Several series have demonstrated that SDD was feasible in various countries, in Europe and North America [10,11,12,13]. However, most of them were single-center series without multi-institutional validation or comparisons between inpatient and outpatient surgeries. Two series have suggested that continence recovery after RARP could be impacted by the type of hospitalization with improved outcomes after SDD [8,11]. Nevertheless, continence recovery after RARP also depends on various parameters, such as nerve-sparing procedures, surgeon experience, patient age, which were not taken into account in these comparative series. Moreover, patient selection for SDD peri-operative management may highly vary among centers regarding ERAS, prehabilitation and follow-up protocols. All of these factors may impact RARP outcomes.

Here, we report post-operative and >6-month functional outcomes after RARP by comparing SDD versus inpatient surgery in a homogenous, multi-surgeon, single-center cohort of consecutive patients who all were eligible for SDD based on our center criteria (distance home-hospital <150 km, no oral anticoagulation).

## 2. Materials and Methods

We included patients undergoing RARP (*n* = 376) since the introduction of the SDD program (March 2018–March 2020). The study protocol was approved by the local institutional ethics committee. An already developed ERAS protocol was applied to all patients since 2016. For this analysis, we only included patients eligible for SDD according to our local criteria (no oral anticoagulation; distance home-hospital <150 km) and having a >6 months post-operative follow-up.

RARP was performed by four experienced surgeons, all beyond their learning curve, having performed more than 200 procedures at study entry. Flow chart is shown in Figure 1. All patients were continent before RARP. No variation in surgical technique (standard transperitoneal approach, nerve-sparing, apex reconstruction, extent of lymph node dissection) was noted per surgeon during the study period. Postoperative course was standardized in terms of care. No drain tube was placed. Bladder catheter was removed at day 7. Postoperative visits were scheduled at month 1 and 6. In case of persistent urinary incontinence defined as >2 daily pads at 1 month after surgery, 20 sessions (2/week) of physiotherapist-guided pelvic floor muscle training were prescribed. The minimal post-operative follow-up was 6 months for all patients. Discharge at home was programmed at day 1 in inpatient patients.

The SDD program has already been described [11]. Briefly, the postoperative course contained liquid fluid intake 2 h after surgery, solid intake at 4 h after surgery, stand up and walk with the physiotherapist 4 h after surgery, oral route without any opioid for analgesia.

Oral level 1 analgesics were prescribed on demand with the possibility of level 2 drugs if needed. The first dose of heparin for thromboprophylaxis was injected subcutaneously by the hospital nurse before discharge. All patients were discharged at home. A 24 h post-discharge phone call was systematically given by a nurse.

Data were collected prospectively and included the following items: patient age, body mass index (BMI), American Society of Anesthesiologists (ASA) score, Charlson comorbidity index, medical treatments, oncologic data, date and duration of surgery, blood loss, length of stay, 90-day readmission rate (including readmissions in emergency and/or urology departments), unplanned visits (including readmissions and unplanned urology visits), continence status at 1 and 6 months. Readmission events were double-checked with the Programme Médicalisé des Systèmes d’Informations (PMSI) data. We used a strict definition of continence recovery as the absence of any pad (no safety pad). Perioperative complications were reported according to the Clavien–Dindo classification.

Endpoints were the continence recovery (strictly defined as no safety pad at 1 and 6 months) and the perioperative parameters (blood loss, operative time, length of stay, transfusion, complication, and readmission rates). Comparisons were made between SDD and inpatient RARP in univariable models. Parameters were compared using 2-tailed tests as appropriate. The limit of statistical significance was defined as *p* < 0.05. SPSS 22.0 software (SPSS, Inc., Chicago, IL, USA) was used for analysis.

## 3. Results

### 3.1. Overall Cohort (n = 180)

Median patient age was 66.7 years. Median duration of surgery (from skin to skin) was 134 min (interquartile range (IQR): 118–152). Median BMI and Prostate Specific Antigen (PSA) were 26 kg/m^2^ and 7.0 ng/mL, respectively. Median blood loss was 200 mL. Lymph node dissection and nerve-sparing procedures were performed in 76.7% and 82.2% of cases, respectively. No conversion to open surgery was noted. Median length of stay was 1.0 day. Transfusion, 90-day readmission, and 90-day complication rates were 0.6%, 9.4%, and 26.1%, respectively. Grade 1–2 and grade 3 complication rates were 21.6% and 4.4%, respectively. No grade 4–5 complications occurred. The 1- and 6-month continence rates (no safety pad) were 51.1% and 81.1%, respectively. Median follow-up was 19.5 months (Table 1).

### 3.2. SDD versus Inpatient RARP

Outcomes of 42 SDD patients were compared with those of 138 inpatient patients (Table 2). No difference was seen regarding patient features (all *p* values >0.05; age, BMI, ASA score, PSA, prostate volume). The proportion of SDD RARP was stable over time (23.0% and 23.7% in 2018–2019 and 2019–2020). During the study period, the proportion of SDD cases ranged from 4.8% to 37.8% per surgeon. Surgery techniques were comparable among groups (lymph node dissection, nerve-sparing surgery). Peri-operative outcomes did not differ between groups (operative time, blood loss). Pathology parameters concerning International Society of Urological Pathology (ISUP) grade (*p* = 0.332), pT stage (*p* = 0.603), positive margins (*p* = 0.384) and pN status (*p* = 0.573) were comparable between SDD and inpatient surgery.

The 90-day unplanned visits, readmission and complication rates were 9.5%, 7.1%, and 19.0% in SDD patients versus 14.5% (*p* = 0.407), 10.1% (*p* = 0.560), 28.3% (*p* = 0.234) in inpatient RARP, respectively. Trends favoring SDD were not statistically significant. Continence rates at 1- and 6-months were comparable between SDD and inpatient RARP: 54.8% versus 50.0%, *p* = 0.589; 83.3% and 80.4%, *p* = 0.674).

## 4. Discussion

The advent of minimally-invasive RARP including the development of robotic surgery has led to a dramatical reduction of the length of stay over time [1,10,11,12]. However, this evolution has inversely followed the continuous increase in healthcare costs, including the costs of surgical devices, which has induced economic pressure from private insurances and public health care systems to shorten hospitalizations with the goal of costs reduction. Early discharge directly lightens hospitalization costs and could generate indirect benefits at a larger level by accelerating return to work and to normal physical activity. These considerations have encouraged urologists and structures to promote SDD RARP. However, to date, few series assessing the short-term safety of SDD RARP have been published. The vast majority of studies were single-center, and even single-surgeon series, without long-term follow-up and functional recovery outcomes [10,11,12,14]. The only multi-institutional study has recently provided interesting data confirming the safety of SDD in different centers, involving different surgeons, different local criteria, and different peri-operative pathways, such as ERAS and prehabilitation programs [15].

Moreover, few comparative studies assessing the safety of SDD versus contemporary inpatient surgery have been reported. Abaza et al. recently reported outcomes from 500 patients undergoing RARP plus lymph node dissection [14]. Patients with SDD (49.2% of the overall cohort) were compared with inpatient patients. No increase in complications, readmissions, or unplanned visits was associated with SDD. Ploussard et al. confirmed the comparable short-term peri-operative outcomes between the two types of hospitalization [13]. Khalil et al. analyzed data from the National Surgical Quality Improvement Program of the American College of Surgeons by comparing SDD patients (*n* = 258) and overnight stay patients (*n* = 1290) [16]. Overall morbidity, including readmission and reoperation rates, were low and similar in both groups.

A couple of strategies in order to facilitate SDD have been described in literature. In a large retrospective review, Ferroni and Abaza reported a decreased mean length of stay (0.57 vs. 1.00 days, *p* < 0.001) and decreased 12 h postoperative pain in patients undergoing RARP with low pneumoperitoneum pressures (6 mmHg) when comparing with standard CO_2_ pressures (15 mmHg), with just slightly longer operative time and higher blood loss (10.5 min longer, 20 mL higher, respectively) [17]. In addition, Shahait et al. in their retrospective analysis advocate for the robot assisted transversus abdominis pain bloc (TAP), showing that it was associated with decreased 6–12 h and 12–18 h postoperative pain and reduced analgesia requirements (*p* < 0.05), thus potentially facilitating SDD after RARP [18].

Given that SDD seems to be at least as safe as inpatient surgery, SDD could also lead to significant cost reductions. Abaza et al., in a U.S. healthcare system suggested a not negligible reduction in charges per patient [14]. We also denoted same benefits (10% reduction) in the French healthcare system [13].

The patient education and preparation prior to RARP play a key role in patient perception and acceptance of SDD. Indeed, the patient willingness to undergo SDD may represent potential barriers. Dobbs et al. suggested that only one-third of patients felt ready to be discharged on the day of their surgery [19]. The insufficient education about catheter care was one of the main barriers limiting their adherence to SDD. In line with these findings, an initial feasibility study in France has not encouraged SDD concluding that only 1% of RARP patients were suitable for inpatient surgery based on the post-operative Chung score at day 0 [20]. However, the peri-operative protocols regarding patient education, anesthesia, and post-operative care were not adapted to SDD in that series. Conversely, recent series have demonstrated that the implementation of ERAS and prehabilitation pathways might help improve RARP outcomes and promote SDD adoption for patients [4,13].

In the present study, we evaluated both perioperative outcomes and mid-term continence recovery. This second endpoint has not been well addressed in previous studies although two previous reports have suggested a not significant benefit from SDD concerning early continence recovery after RARP [10,13]. Our aim was to confirm or not this trend in a larger cohort of patients operated on by multiple surgeons and with a minimal 6-month follow-up. We also integrated several factors that might impact urinary continence recovery. Based on this comparative analysis, we did not confirm clinically meaningful differences regarding the 1- and 6-month continence recovery rates between both groups.

The main limitation of SDD series remains the selection bias given the lack of randomization. Thus, SDD patients may be more actively motivated to recover rapidly after RARP than overnight stay patients. We tried to reduce this potential bias by including in the inpatient surgery cohort only men who were eligible for SDD based on our local criteria (distance, anticoagulation). Despite the fact that all pre-operative parameters (including age, ASA score, BMI) were comparable in SDD versus inpatient cohorts in our patient population, the strict equivalence between groups cannot be guaranteed. Another point was that the acceptance of SDD may be gradual as the surgeons were becoming more and more comfortable with offering SDD once they have observed the safety of SDD. Thus, in our experience, whereas the overall proportion of SDD RARP was comparable over time (one-fourth), we noted surgeon-based differences concerning the proportion of SDD cases with a proportion of SDD ranging from 5% to 40% among the four surgeons involved during the study period. We also noted that the SDD rate per surgeon continuously increased over time (to reach 60% for one surgeon) despite no change in protocol or surgery scheduling.

Taken together, our findings confirm that routine SDD following RARP can be safely offered without increasing readmissions or emergency visits. However, no clear benefit favoring SDD has been demonstrated although a trend towards less complications and readmission was observed with similar functional recovery. It seems reasonable to let the patient self-select the type of hospital without imposing SDD.

Our study is not devoid of limitations. First, the psychology of patients has not been assessed prior to RARP. The anxiety of patients may have an important impact on the choice of SDD and on the delay to recover after surgery. Quality of life after RARP has not been well compared between SDD and inpatient surgery. Nevertheless, Bajpai et al. demonstrated that SDD had a positive impact on pain, general activity, and perceived overall health two days after discharge compared with next day discharge [21]. Second, no randomization was done, which may introduce selection biases between SDD and inpatient cases although the exclusion of inpatient surgeries in men who did not fulfill the SDD criteria reduced that risk. Third, the single-center design may also limit the generalization of our results, even if we analyze patient outcomes in a 4-surgeon cohort. However, the multi-institutional proof of the SDD safety has been already demonstrated [15].

## 5. Conclusions

This multi-surgeon comparative study confirms the safety of routine SDD RARP in terms of perioperative and functional outcomes. SDD globally represents one fifth of this 4-surgeon cohort. When comparing to inpatient cases also eligible for SDD, no difference regarding the 1- and 6-month strict continence rates was observed. Trends favoring SDD in terms of complications, emergency visits, and readmission have to be confirmed.

## Figures and Tables

**Figure 1 jcm-10-00661-f001:**
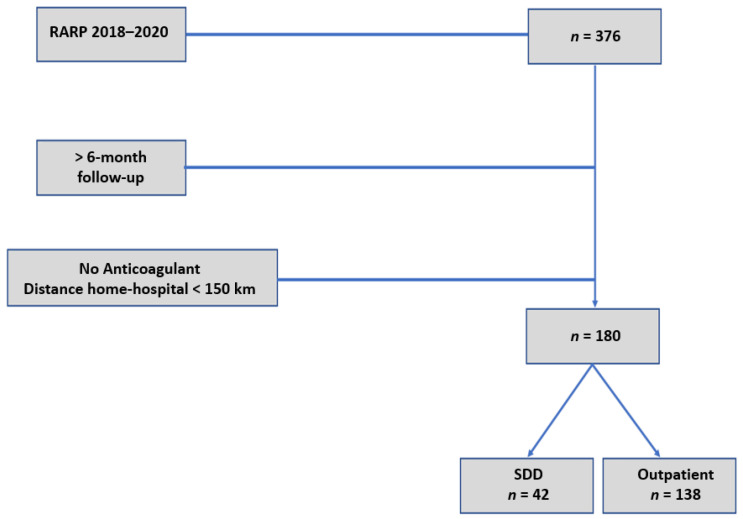
Flowchart of study design. RARP: robot-assisted radical prostatectomy; SDD: same day discharge.

**Table 1 jcm-10-00661-t001:** Overall cohort clinical, perioperative and pathological characteristics.

	Overall Cohort *n* = 180
Mean age, years	65.7
(median, IQR)	66.7 (61.5–70.5)
Mean BMI, Kg/m^2^	26.3
(median, IQR)	26.0 (24.1–28.4)
ASA score:	
2	85 (47.2%)
3	1 (0.6%)
Mean Charlson comorbidity index	4.2
median, IQR)	4.0 (4–5)
Year of RARP	
2018–2019	*n* = 87
2019–2020	*n* = 93
Mean PSA, ng/mL	8.4
(median, IQR)	7.0 (5.3–9.4)
Mean prostate volume, mL	51.4
(median, IQR)	47.0 (35–61)
Mean operative time, min	137
(median, IQR)	134 (118–152)
Mean blood loss, mL	233
(median, IQR)	200 (100–300)
Pelvic lymph node dissection	138 (76.7%)
Nerve-sparing surgery	148 (82.2%)
Mean length of stay, days	1.2
(median, IQR)	1.0 (0–2)
ERAS pathway	180 (100%)
Transfusion	1 (0.6%)
Unplanned visits	24 (13.3%)
Readmission	17 (9.4%)
Clavien–Dindo grade:	
1	17 (9.4%)
2	22 (12.2%)
3	8 (4.4%)
Pathological Grade:	
1	7 (3.9%)
2–3	150 (83.3%)
4–5	23 (12.8%)
pT stage:	
pT2	92 (51.1%)
pT3a	75 (41.7%)
pT3b	13 (7.2%)
Positive surgical margins	57 (31.7%)
pN1 status:	7 (5.0%)
No safety pad at 1 month	92 (51.1%)
No safety pad at 6 month	146 (81.1%)
Mean follow-up, months	19.2
(median, IQR)	19.5 (10.6–27.2)

Abbreviations: IQR—interquartile range; ASA—American Society of Anesthesiologists; RARP—Robot assisted radical prostatectomy; PSA—Prostate Specific Antigen; ERAS—enhanced recovery after surgery.

**Table 2 jcm-10-00661-t002:** Comparison between same day discharge (SDD) and inpatient robot-assisted radical prostatectomy (RARP) cohorts.

	SDD *n* = 42	Inpatient *n* = 138	*p* Value
Mean age, years	65.5	65.8	0.726
(median, IQR)	66.1 (60.5–69.6)	64.3 (62.7–71.9)	
Mean BMI, Kg/m^2^	26.1	26.4	0.521
(median, IQR)	24 (19.3–28.5)	25.2 (23.4–29.1)	
ASA score:			0.291
2	25 (59.5%)	60 (43.5%)	
3	0	1 (0.7%)	
Mean Charlson comorbidity index	4.1	4.2	0.122
median, IQR)	3.3 (2.9–6.2)	3.4 (3.1–6.5)	
Year of RARP			0.916
2018–2019	23.0%	-	
2019–2020	23.7%	-	
Mean PSA, ng/mL	10.0	8.0	0.081
(median, IQR)	8.4 (6.7–15.2)	7.1 (5.8–12.9)	
Mean prostate volume, mL	49.3	52.0	0.477
(median, IQR)	50.1 (42.6–53.7)	50.3 (47.3–58.6)	
Mean operative time, min	135	137	0.521
(median, IQR)	129.6 (125.5–148.3)	131.2 (128.7–150.6)	
Mean blood loss, mL	230	234	0.886
(median, IQR)	227.5 (210.4–250.6)	230.7 (225.7–264.1)	
Pelvic lymph node dissection	32 (76.2%)	106 (76.8%)	0.934
Nerve-sparing surgery	33 (78.6%)	115 (83.3%)	0.480
Mean length of stay, days		1.6	<0.001
(median, IQR)		1 (1.2–2.3)	
ERAS pathway	42 (100%)	138 (100%)	1.00
Transfusion	0	1 (0.7%)	0.580
Unplanned visits	4 (9.5%)	20 (14.5%)	0.407
Readmission	3 (7.1%)	14 (10.1%)	0.560
Complication	8 (19.0%)	39 (28.3%)	0.234
Clavien–Dindo grade:			0.538
1	2 (4.8%)	15 (10.9%)	
2	5 (11.9%)	17 (12.3%)	
3	1 (2.4%)	7 (5.1%)	
Pathological Grade:			0.332
1	2 (4.8%)	5 (3.6%)	
2–3	34 (81.0%)	116 (84.1%)	
4–5	6 (14.3%)	17 (12.3%)	
pT stage:			0.603
pT2	23 (54.8%)	69 (50.0%)	
pT3a	15 (35.7%)	60 (43.5%)	
pT3b	4 (9.5%)	9 (6.5%)	
Positive surgical margins	31 (26.2%)	46 (33.3%)	0.384
pN1 status:	1 (3.1%)	6 (5.6%)	0.573
No safety pad at 1 month	23 (54.8%)	69 (50.0%)	0.589
No safety pad at 6 month	35 (83.3%)	111 (80.4%)	0.674
Follow-up, months	18.4	19.4	0.492
(median, IQR)			

## Data Availability

No new data were created or analyzed in this study. Data sharing is not applicable to this article.

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
