# Peer review of "Same Day Discharge versus Inpatient Surgery for Robot-Assisted Radical Prostatectomy: A Comparative Study"

_jcm, 2021, doi:10.3390/jcm10040661_

Round 1
Reviewer 1 Report
Title:
The title reflects the aims of the study well.
Abstract:
The abstract reflects their work and the aims.
Introduction:
Introduction is well-written and easy to follow.
There are some points that I would suggest.
Authors could mention the merit of RAL-IVCT compared with open IVCT.
Methods:
Overall, the study design is clear and appropriate for the aims of the study.
Results:
The results were clearly presented and could be followed easily in figure and tables.
There are some points that I would suggest.
・In table 1, I suggested that authors remove or clarify the unclear data (average?). (i.e. age 65.7, BMI 26.3)
・In table 2, I suggested that authors added the IQR.
Discussion:
The discussion addressed the research question and interpreted the results in a plausible way.
Conclusions:
The conclusion reflects their work in a plausible way.
Author Response
Response to Reviewer 1 Comments
Dear reviewer,
Thank you so much for your work and your constructive comments.
We have duly considered your suggestions.
Our revisions are listed in this Letter.
Point 1: Introduction: Authors could mention the merit of RAL-IVCT compared with open IVCT.
Response 1: In our study we only included patients that underwent radical prostatectomy by robot-assisted laparoscopic approach. We do acknowledge the merit of RAL-IVCT compared with the open approach. However, taken into account that in our study we referred only to radical prostatectomy as surgical treatment for prostate cancer, we believe that it is not of great interest for our endpoints and message.
Point 2: Results: In table 1, I suggested that authors remove or clarify the unclear data (average?). (i.e. age 65.7, BMI 26.3). In table 2, I suggested that authors added the IQR.
Response 2: This has been done for all suitable variables; we were referring to the average (mean) value. We have added the IQR values for table 2.

Reviewer 2 Report
Same Day Discharge versus Inpatient Surgery for Robot-Assisted Radical Prostatectomy: a Comparative Study
Abstract:
I’d include the sample size in the abstract (especially in light that the trends regarding complications, ER visits and readmission trended towards significance but didn’t reach it)
Introduction:
Nice introduction – agreed that most available studies have been pragmatic rather than comparative.
Discussion:
I'd include a brief discussion of strategies investigators have used to facilitate SDD such as reducing the pneumoperitoneum pressure (Abaza et al) or Robotic TAP block (Shahait et al "robot-assisted transversus abdominis plan block: description of the technique and comparative analysis")
In the limitations, could note that specific patient populations (BMI = 26) may not be generalizable
Author Response
Response to Reviewer 2 Comments
Dear reviewer,
Thank you so much for your work and your constructive comments.
We have duly considered your suggestions.
Our revisions are listed in this Letter.
Point 1: Abstract: I’d include the sample size in the abstract (especially in light that the trends regarding complications, ER visits and readmission trended towards significance but didn’t reach it)
Response 1: We included the sample size for all cohorts in abstract.
Point 2: Discussion: I'd include a brief discussion of strategies investigators have used to facilitate SDD such as reducing the pneumoperitoneum pressure (Abaza et al) or Robotic TAP block (Shahait et al "robot-assisted transversus abdominis plan block: description of the technique and comparative analysis")
In the limitations, could note that specific patient populations (BMI = 26) may not be generalizable.
Response 2: We included the 2 references in our Discussion section : A couple of strategies in order to facilitate SDD have been described in literature. In a large retrospective review, Ferroni and Abaza reported a decreased mean length of stay (0.57 vs 1.00 days, p<0.001) and decreased 12h postoperative pain in patients un-dergoing RARP with low pneumoperitoneum pressures (6mmHg) when comparing with standard CO2 pressures (15 mmHg), with just slightly longer operative time and higher blood loss (10.5 min longer, 20 ml higher, respectively) [20]. In addition, Shahait et al. in their retrospective analysis advocate for the robot assisted transversus ab-dominis pain bloc (TAP), showing that it was associated with decreased 6-12h and 12-18h postoperative pain and reduced analgesia requirements (p<0.05), thus potentially facilitating SDD after RARP [21].

Reviewer 3 Report
A number of issues with the manuscript as outlined below:
- Table 2 refers to "Outpatient" yet the rest of the article refers to "inpatient". Please correct
- Line 198 "Our study is not devoted of limitations" - I assume you mean devoid?The use of the word devoted is not appropriate here
- Small sample size in the SDD arm. The numbers are skewed against SDD which raises the question of serious selection bias. Can the author clearly outline how the decision for SDD was made. e.g Was it pre op vs intra op vs post op, were the patients pre-counselled
- Selection bias is a very big issue with the study design and data presented here which may explain why the numbers are lower in the SDD arm. Can the authors also clarify if there were other differences between the 2 groups apart from the fact that one group went home same day while the other group stayed as inpatient? e.g I assume the SDD had no drain tube so it would be good to know if the inpatient group also had no drain tubes for example
Author Response
Response to Reviewer 3 Comments
Dear reviewer,
Thank you so much for your work and your constructive comments.
We have duly considered your suggestions.
Our revisions are listed in this Letter.
Point 1: Table 2 refers to "Outpatient" yet the rest of the article refers to "inpatient". Please correct
Response 1: This has been corrected, it refers indeed to the inpatient cohort.
Point 2: Line 198 "Our study is not devoted of limitations" - I assume you mean devoid?The use of the word devoted is not appropriate here
Response 2: This has been corrected, it refers indeed to the fact that our study is not without limitations. The error is though at line 211, not 198.
Point 3: Small sample size in the SDD arm. The numbers are skewed against SDD which raises the question of serious selection bias. Can the author clearly outline how the decision for SDD was made. e.g Was it pre op vs intra op vs post op, were the patients pre-counselled.
Response 3: We believe that this aspect has been clearly defined in the Materials and Methods section: ‘For this analysis, we only included patients eligible for SDD according to our local cri-teria (no oral anticoagulation; distance home-hospital <150 km)’- line 63-64. SDD patients were selected if they fulfilled the 2 above mentioned criteria.
Point 4: Selection bias is a very big issue with the study design and data presented here which may explain why the numbers are lower in the SDD arm. Can the authors also clarify if there were other differences between the 2 groups apart from the fact that one group went home same day while the other group stayed as inpatient? e.g I assume the SDD had no drain tube so it would be good to know if the inpatient group also had no drain tubes for example.
Response 4: We acknowledge that the sample size for the SDD arm is quite small although the overall cohort (n=180) we believe to be adequate. However, after applying the criteria for selecting SDD patients (no oral anticoagulation; distance home-hospital <150 km), we were left with 42 patients for SDD vs 138 patients for Inpatient RARP. Apart from the numbers of the 2 cohorts, they were similar in terms of clinical, perioperative and oncological outcomes. The decision of leaving or not a drain tube was depending on the intraoperative blood loss (>400-500 ml), which was exceptional (more than 90% of patients in Inpatient RARP did not have a drain tube placed) and was removed in day 1 postop before discharge.

Round 2
Reviewer 3 Report
Thank you for the above revisions
However, please clarify the following which still has not been addressed:
- Can the author clearly outline how the decision for SDD was made. e.g Was it pre op vs intra op vs post op, were the patients pre-counselled. The worry is regarding selection bias for the cases that eventually progressed to SDD. All 180 cases analyzed met your criteria (thus were eligible for SDD) but it is not clear in your methods how the decision to proceed to SDD was made for the 42 cases. Otherwise it appears as though these were carefully selected cases since the other 138 cases also met your criteria yet stayed as inpatient which may be a significant selection bias issue. Simply put, if all 180 cases met your criteria, why only 42 cases? Surgeon choice? Intraop concerns? Pt preference? Were these cases you knew would definitely do well with SDD?
- Would be worth mentioning in you manuscript that 90% of the inpatient group also didn't have a drain tube (as eluded to in your response letter) as this adds to to your argument in favour of SDD.
Author Response
Response to Reviewer 3 Comments
Dear reviewer,
Thank you again for your work and your constructive comments.
We have duly considered your suggestions.
Our revisions are listed in this Letter.
Point 1: Can the author clearly outline how the decision for SDD was made. e.g Was it pre op vs intra op vs post op, were the patients pre-counselled. The worry is regarding selection bias for the cases that eventually progressed to SDD. All 180 cases analyzed met your criteria (thus were eligible for SDD) but it is not clear in your methods how the decision to proceed to SDD was made for the 42 cases. Otherwise it appears as though these were carefully selected cases since the other 138 cases also met your criteria yet stayed as inpatient which may be a significant selection bias issue. Simply put, if all 180 cases met your criteria, why only 42 cases? Surgeon choice? Intraop concerns? Pt preference? Were these cases you knew would definitely do well with SDD?
Response 1: Indeed, all 180 cases were suitable for SDD. Only 42 patients were finally included in the SDD cohort because of patient preference. Some presented anxiety when declared fit to be discharged, some described persistent nausea and intense fatigue and some described important postoperative pain, reasons for which all these patients (the remaining 138) were admitted for 1 night on hospitalisation. Of note that all of the remaining 138 patients were discharged the following day, without any signs of complications.
Point 2: Would be worth mentioning in you manuscript that 90% of the inpatient group also didn't have a drain tube (as eluded to in your response letter) as this adds to to your argument in favour of SDD.
Response 2: I apologise for the confusion, as a matter of fact none of the patients (n=180) included in our study had a drain tube placed. The previous mentioned percentage was referring to the rest of cohort (n=376) and to the patients that were not finally included in our study. We have corrected this in the manuscript (line 71).
